# Pine pollen reverses the function of hepatocellular carcinoma by inhibiting α-Enolase mediated PI3K/AKT signaling pathway

**Yanhong Luo**[1☯], **Chun Guo**[1☯], **Caixia Ling**[1], **Wenjun Yu**[3], **Yuanhong Chen**[4], **Lihe Jiang**[4], **Qiuxiang Luo**[1], **Chunfang Wang**[5]*, **Weixin Xu**[2]*

1 Department of Laboratory Medicine Science, Youjiang Medical University for Nationalities, Baise, China, 2 Department of Pharmacy, Yancheng Clinical College of Xuzhou Medical University, The First People's Hospital of Yancheng, Yancheng, China, 3 Department of Mechanical Engineering and Materials Science, Duke University, Durham, NC, United States of America, 4 School of Basic Medicine, Youjiang Medical University for Nationalities, Baise, China, 5 Key Laboratory of Researcmon Clinical Molecular Diagnosis for High Incidence Diseases in Western Guangxi of Guangxi Higher Education Institutions, Baise, China

☯ These authors contributed equally to this work.
* 872348531@qq.com (WX); 578457776@qq.com (CW)

**Data Availability Statement:** The data generated in this study is included in the manuscript itself and uploaded as supplementary information.

## Abstract

### Objective

This study aimed to investigate the influence of pine pollen (PP) on hepatocellular carcinoma (HCC) behavior *in vitro* and *in vivo* and explore its mechanism of action by focusing on the phosphatidylinositol 3-kinase/protein serine-threonine kinase (PI3K/AKT) signaling pathway and α-Enolase (*ENO1*) gene expression.

### Methods

We performed a bioinformatics analysis of *ENO1*. HCC cells overexpressing *ENO1* were developed by lentivirus transfection. Cell proliferation, invasion, and migration were assessed using the cell cytotoxicity kit-8 assay, transwell assay, cell scratch test, and *ENO1* inhibiting proliferation experiment. Protein expression was analyzed using Western blot. The *in vivo* effects of PP on HCC xenografts were also assessed in mice. The serum of nude mice in each group was analyzed for alanine aminotransferase (ALT), aspartate aminotransferase (AST), and AST/ALT. The tumor blocks of nude mice were weighed, and proteins were extracted for Western blot.

### Results

Compared to normal cells, the phosphorylation of *ENO1* at the S27 site was most significant in HCC cells and was closely related to cell proliferation. *In vitro*, the PP solution inhibited the proliferation, invasion, and migration of *ENO1* overexpressing cells compared with empty-vector-transfected cells. In mice bearing HCC, PP injection inhibited the overexpression of *ENO1*, affected serum ALT, AST, and AST/ALT levels, and reduced tumor weight.

**Funding:** This study was supported by the National Natural Science Foundation of China (No. 81960303); Project of Guangxi Key Laboratory of Molecular Pathology of Hepatobiliary Diseases (No. [2021]61); the Foundation of Modern Industrial College of Biomedicine and Great Health, Youjiang Medical University for Nationalities, Baise, Guangxi, China; School level project of Youjiang Medical University for Nationalities (No. yy2021sk012).

**Competing interests:** The authors confirm that they have no competing conflicts of interest.

**Abbreviations:** HCC, hepatocellular carcinoma; PP, pollen pini; PS, physiological saline; IC50, half maximal inhibitory concentration; ENO1, α-Enolase; PI3K / AKT, phosphatidylinositol 3-kinase / protein serine-threonine kinase; ALT, alanine aminotransferase; AST, aspartate aminotransferase; MBP-1, C-MYC promoter binding protein - 1; RT-qPCR, real-time fluorescent quantitative polymerase chain reaction; BCA, polybutylcyanoacrylate; SDS-PAGE, sodium dodecyl sulfate-polyacrylamide; NC, electrotransferred to the nitric acid fiber; TBST, Tris buffed saline tween-20; GAPDH, glyceraldehyde-3-phosphate dehydrogenase; BRCA, breast cancer; KIRC, kidney clear cell carcinoma; HNSC, head and neck squamous carcinoma; CC, colon cancer; GBM, glioblastoma multiform; LIHC, liver hepatocellular carcinoma; LUAD, lung adenocarcinoma; OV, ovarian cancer; PAAD, pancreatic adenocarcinoma; UCES, Uterine corpus endometrial carcinoma; ESCA, esophageal cancer; THCA, thyroid cancer; GPI, glucose-6-phosphate isomerase; PGAM1, phosphoglycerate mutase 1; ALDOA, fructose bisphosphate aldolase A; PGK1, phosphoglycerate kinase 1; HIF-1, hypoxia-inducible factor-1; ERBB2, Erb-B2 receptor tyrosine kinase 2; EIF2α, eukaryotic translation initiation factor2α; MMP2, matrix metallopeptidase 2; 2PG, 2-phosphoglyceric acid; PEP, phosphoenolpyruvic acid; VEGF, vascular endothelial growth factor; EMT, epithelial-mesenchymal transition; ERs, endoplasmic reticulum stress.

However, the expression of proliferation-related proteins in tumors overexpressing *ENO1* was higher than in empty transfected tumors.

## Conclusion

PP inhibits HCC by regulating the expression of *ENO1* and *MBP-1* and suppressing the PI3K/AKT pathway by inhibiting C-MYC and erb-B2 receptor tyrosine kinase 2.

## 1. Introduction

Hepatocellular carcinoma (HCC) is the most frequent type of primary liver cancer and the fourth leading cause of cancer-related mortality worldwide [1]. Surgery (such as resection and liver transplantation) and systemic therapy are the primary treatment methods for HCC. However, the overall resectability rate for HCC is only 10%–25%, while systemic therapy has been associated with certain side effects [2]. Therefore, the search for an alternative and safer therapy is urgently required.

Pine pollen (PP) is a traditional Chinese medicine that contains various trace elements and nutrients essential to the human body. PP has been used for decades for various health-related purposes, such as treating various conditions, including constipation, colds, and prostate disease, supplementing the diet or adding to foods, slowing aging, reducing fatigue, and boosting testosterone levels. PP has also been used to treat cancer [3]. For instance, PP can inhibit cell proliferation and enhance apoptosis in human liver cancer cells [4]. However, the underlying molecular mechanisms are still not fully understood.

α-Enolase (ENO1) is a plasminogen receptor expressed on the cell surface, particularly in malignant cells, such as HCC. Studies have revealed that *ENO1* promotes the occurrence and metastasis of pancreatic and breast cancers by activating (PI3K/AKT) pathway and affecting the glycolytic pathway [5, 6].

In this study, we assessed the effect of PP on HCC behavior and explored its mechanism of action by focusing on *ENO1* gene expression and the PI3K/AKT signaling pathway.

## 2. Materials and methods

### 2.1. Ethics approval and consent to participate

This study was conducted in accordance with the principles of the Helsinki Declaration. The use of nude mice in the animal experiment in this study was supervised by the experimental animal ethics committee of Youjiang Medical College for Nationalities and passed the ethical review with the ethical review number of 2023022401. The cell line used is SMMC-7721 purchased from Wuhan Boster Company, with a quantity of 1 and a batch number of 20170310–04.

### 2.2. The protein phosphorylation level of ENO1 in pan-cancer

The UALCAN platform and the CPTAC database were used to compare the phosphorylation levels of ENO1 in primary tumors and normal tissues.

### 2.3. ENO1 immune infiltration in pan cancer

To explore the relationship between *ENO1* expression and immune infiltration level, TIMER2.0 (http://timer.cistrome.org/) was used, and the "Gene" module of the platform "Immune" was input with the "*ENO1*" gene.

## 2.4. Enrichment analysis of ENO1-related genes

The experimental data of 50 ENO1 binding proteins was obtained using the STRING website (https://string-db.org/), and "ENO1" was input in the "Protein by name" section of "HOMO sapiens." The main parameters were set as follows: network type: full STRING network; measurement of network edges: evidence; active interaction sources: experiments; minimum required interaction score: low confidence (0.150); maximum number of contactors to display: no more than 50 contractors. The top 100 targeted genes associated with *ENO1* expression were identified using GEPIA2 (http://gepia2.cancer-pku.cn/#analysis) and the platform's "Similar Genes Detection" module. Jvenn, an interactive Venn diagram viewer, was used to compare ENO1 binding genes and interacting genes and analyze the intersection of the aforementioned 100 genes and 50 interacting proteins. The common intersection genes and the *ENO1* gene were evaluated for paired gene correlation in the GEPIA2 "correlation analysis" module. The scatter diagram was log2TPM, and the *P* value and correlation coefficient R were given. Next, a heatmap was plotted for the selected common genes using the "Exploration" module of "TIMER2.0". Furthermore, the above 100 genes and 50 interacting proteins were used together in DAVID (https://david.ncifcrf.gov/) to obtain biological process (BP) and molecular function (MF) data. Finally, WeChat (http://www.bioinformatics.com.cn/) was used to conduct GO/KEGG pathway analysis and obtain relevant bubble charts and histograms.

## 2.5. Cell culture

Human SMMC-7721 hepatoma cell line (procured from Wuhan Boster) was continuously cultured in RPMI 1640 culture medium (acquired from Beijing Langeker White Shark Technology) containing 100 mL/L fetal bovine serum (FBS; product No. 11011–8611, procured from Hangzhou Tianhang Sijiqing) and 1% penicillin and streptomycin (product No. p1400-100, obtained from Beijing Solebo Biological) in a saturated moisture content atmosphere containing 5% $CO_2$/95% air at 37˚C. The cells were passaged every 48 h.

## 2.6. Preparation of stably transfected hepatoma cell line and real-time-quantitative polymerase chain reaction (RT-qPCR)

According to the transfection instructions for overexpressed *ENO1* lentivirus (Shanghai Jikai Gene Company), the overexpressed *ENO1* vector carrying green fluorescent protein and empty vector was transferred into the corresponding SMMC-7721 cells, and the transfection was verified by observing the fluorescence under a fluorescence microscope. The stable strain was screened using puromycin, and the expression of *ENO1* was determined using RT-qPCR (Shanghai Yisheng Company). The expression of *ENO1* was analyzed using the $2^{-\Delta\Delta Ct}$ method. The experiment was performed in triplicate, and the average value was calculated. The primer sequences used for RT-qPCR were as follows: F: 5'-gtaccgccacatcgctgacttg-3' and R: 5'-gaaccgccattgatgatgatgaacg-3' for ENO1 and F: 5'-gcaccgtcaaggctaac-3' and R: 5'-tggtgaagagagagagccagtgta-3' for glyceraldehyde-3-phosphate dehydrogenase (GAPDH).

## 2.7. Grouping

Cells were divided into four groups: (1) *ENO1* gene overexpression stable transfection group; (2) *ENO1* overexpression stably transfected cells plus PP solution group; (3) SMMC-7721 cells transfected with empty lentivirus; (4) SMMC-7721 cells transfected with empty lentivirus and treated with PP solution.

## 2.8. Cell cytotoxicity kit-8 (CCK-8) assay

Briefly, cell lines were cultured in a 96-well plate ($5 \times 10^4$ cells/mL in 100 μL). After 24 h, the cells were exposed to PP (1.25, 2.50, 5.00, 7.50, and 10.00 μg/μL) (Yantai New Era Health Industry Co., Ltd.) for 12 h. At each time point, 10 μL of sterile CCK-8 reagent (Wuhan Bode Business) was added to each well, and the cells were cultured for 1 h at 37˚C. The absorbance was measured at 450 nm using a microplate reader (American MD Corporation). The optimal half inhibitory concentration ($IC_{50}$) values were calculated from the linear regression of the plot.

## 2.9. Transwell assay

In the upper compartment, cells ($5 \times 10^4$ cells/mL, 250 μL) were introduced, and 800 μL of RPMI-1640 medium with 10% FBS was added to the lower chamber. After 2 h, the upper chamber was treated with 32.68 μL (50 μg/μL) PP. The control group received the same volume of serum-free RPMI-1640 medium. After 24 h, the culture medium was removed, and the upper chamber cells were removed with a sterile cotton swab. Then, cells in the lower chamber were stained with the Transwell Kit (Beijing Langeker Technology Co., Ltd.) and counted under the PET film using an inverted microscope.

## 2.10. Cell scratch test

Cells were seeded in a 6-well plate ($5 \times 10^5$ cells/mL per well). After reaching 90% confluence, a marker was used to draw a line from the bottom of the dish. Three wounds were scratched in the cells with 100 μL sterile pipette tips perpendicular to the line. The cells were gently washed twice with phosphate-buffered saline to remove floating cells. The cells were then incubated in 2 mL medium containing 1% FBS at 37˚C. Images of the scratches were taken using an optical microscope at 4× magnification at 0, 12, 24, and 48 h of incubation and analyzed using ImageJ software.

## 2.11. Western blot

The cells were lysed according to the manufacturer's instructions, and the total protein content was determined using the polybutylcyanoacrylate kit. Proteins were first electrophoresed using sodium dodecyl sulfate-polyacrylamide gel electrophoresis and then electrotransferred to a nitric acid fiber membrane. The membrane was washed, blocked, and mixed with 1:1000 diluted primary antibodies at 4˚C overnight and then with 1:5000 diluted secondary anti-HRP sheep anti-rabbit and HRP Sheep anti-mouse antibodies (Wuhan Bode Company) at room temperature for 2 h. Gel imaging was performed using GAPDH as an internal reference.

## 2.12. ENO1 inhibiting proliferation experiment

The *ENO1* stable transfected and empty transfected cells were incubated in 96-well plates and mixed with 100 μL of a $3 \times 10^4$ cells/mL cell suspension. After 3 h of culture and adherence, 0.5, 1.5, and 2.0 μM of ENO1 inhibitors (MCE company, United States) were administered to each group. Wells without ENO1 inhibitors were used as the negative control, and wells with 0.2% dimethyl sulfoxide (DMSO; Sigma-Aldrich, United States) were used as the solvent control. Each group was parallel to three wells and cultured in an incubator for 0, 24, and 48 h. After adding 20 μL of the CCK-8 reagent to each well, the plates were incubated at 37˚C for 1 h. The absorbance was measured at 450 nm, and the survival rate of each group was calculated.

## 2.13. Animal experiments

Female BALB/C nude mice (3–4 weeks old) were procured from Guangdong Weitong Lihua Laboratory Animal Company. All animals were housed at a temperature of $22 \pm 1°C$, relative humidity of $50 \pm 1\%$, and light/dark cycle of 12/12 h. This study used 28 nude mice. Blood was collected from the eyeball vein plexus of nude mice and anesthetized with isoflurane inhalation before surgery. The nude mice were weighed and monitored daily, and four nude mice were euthanized with 30% weight loss, inactivity, and breathing difficulty within 24 h (by inhalation of $CO_2$). The remaining nude mice were similarly euthanized immediately after the experiment. All animal studies (including mouse euthanasia procedures) were performed in accordance with the regulations and guidelines for institutional Animal Care of Youjiang Medical College for Nationalities and the AAALAC and IACUC guidelines.

Briefly, 0.2 mL of overexpressing ENO1 stably transfected cells or empty transfected cell suspension ($1 \times 10^7$) was breathed into the armpit of each nude mouse, and the knub was formed after three weeks. PP or physiological saline (PS) injection volume was calculated according to the formula for nude mice with each constituent tumor: drug concentration (mg/mL) = $(50 \times D/5000) \div 50\% \times 10^3$ [D is the clinical dosage: mg/(kg)] and the volume of PP or PS injected per mouse (mL) = $(D \cdot G)/IC_{50}$ [G is the weight of each nude mouse] [7]. The PP solution or PS was intraperitoneally injected every other day for two weeks. Subsequently, the serum of nude mice in each group was collected, and the alanine aspartate aminotransferase (AST), aminotransferase (ALT), and AST/ALT ratios were measured. The tumor was then dissected, weighed, and analyzed using Western blot. The expression of related proteins was measured thrice for each protein.

## 3. Statistical analysis

The Statistical Package for the Social Sciences software (version 24.0) was used to perform statistical analyses. Data are presented as the mean ± standard deviation ($x \pm s$), and a one-way analysis of variance was used to perform between-group comparisons. The test level was set at $\alpha = 0.05$. The scratch test adopts Friedman test and Kruskal Wallis test; RT-qPCR was used to calculate the relative gene expression using the $2^{-\Delta\Delta Ct}$ method. Statistical significance was set at $P < 0.05$.

## 4. Results

### 4.1. Protein phosphorylation level of ENO1 in Pan-cancer

We analyzed the phosphorylation levels of ENO1 in regular and initial carcinoma tissues using the CPTAC database (Fig 1A). Measured in initial tissues, the S27 site of ENO1 exhibits the most types of cancer with different phosphorylation levels in all primary tumor tissues and the most significant difference in the phosphorylation levels of liver cancer cells. The cancer types studied included BRCA, KIRC, glioblastoma multiforme (GBM), head and neck squamous carcinoma, LIHC, LUAD, OV, and PAAD (Fig 1B).

### 4.2. ENO1 immune infiltration in pan cancer

TCGA database was used to assess the correlation between tumor immune cell infiltration and *ENO1* gene expression. ENO1 was related to carcinoma microclimate and positively interacted with cancer-related fibroblast infiltration in BRCA, esophageal cancer (ESCA), GBM, KICH, PAAD, and thyroid cancer (Fig 2).

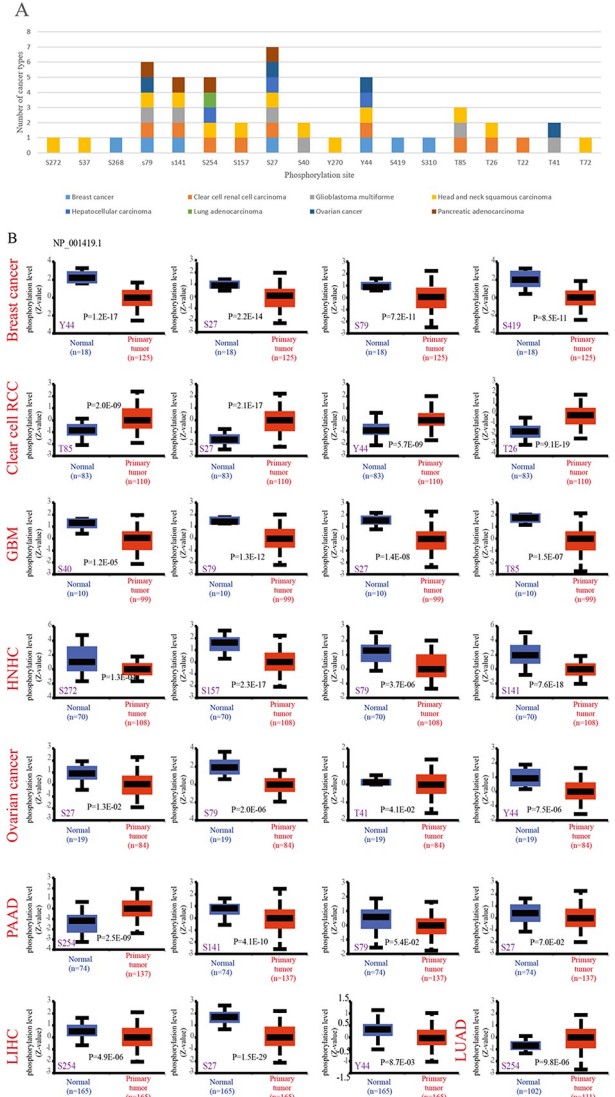

**Fig 1. Phosphorylation analysis of ENO1 protein in different tumors.** (A) Compared with normal tissues, there are the most kinds of cancers with different phosphorylation levels at S27 site of ENO1; (B) Compared with normal tissues, the S27 site of ENO1 had the most significant difference in the phosphorylation level of hepatoma cells (*P* = 1.5e-29).

## 4.3. Improved inquiry of ENO1-related genetic codes

To further analyze the role of *ENO1* in tumorigenesis, we accessed 50 experimental datasets of ENO1 binding proteins and their interaction networks using STRING (Fig 3A). Concurrently, we performed GEPIA2 analysis to identify the top 100 genes related to *ENO1* expression. The expression level of *ENO1* was correlated with glucose-6-phosphate isomerase (*GPI*), phospho-glycerate mutase 1 (*PGAM1*), *LDHA*, *GAPDH*, fructose bisphosphate aldolase A (*ALDOA*), and six genes, including phosphoglycerate kinase 1 (*PGK1*) (all *P* < 0.001) (Fig 3B). Thermo-gram analysis revealed a positive correlation with all six genes mentioned above (Fig 3C). Cross-inquiry of the two datasets revealed nine common genes (Fig 3D).

Subsequently, we analyzed the two datasets using KEGG and GO analyses. KEGG analysis demonstrated that the role of *ENO1* in carcinoma pathogenesis may be associated with carbon metabolism, amino acid biosynthesis, the hypoxia-inducible factor-1 (HIF-1) signaling pathway,

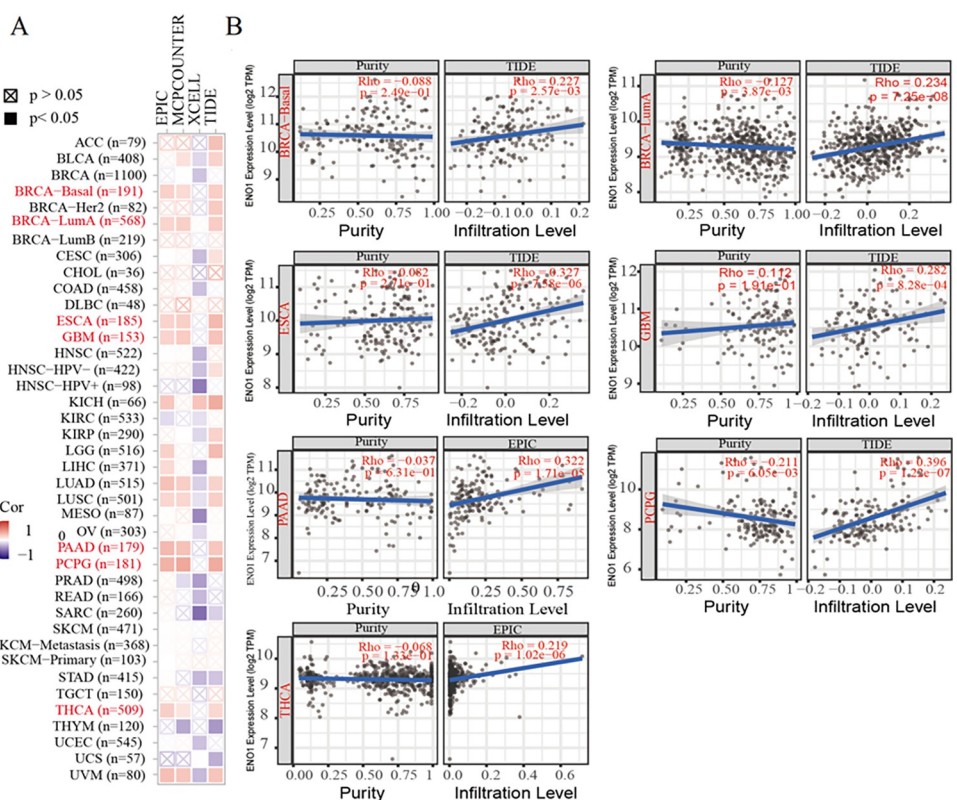

**Fig 2. Correlation between ENO1 expression and immune cell infiltration.** (A) Number of cases of various cancers; (B) In BRCA, esophageal cancer (ESCA), GBM, PAAD and thyroid cancer (THCA), ENO1 was positively correlated with the invasion of cancer-related fibroblasts ($P<0.05$).

the metabolic pathway, central charcoal anabolism, the pentose phosphate pathway, nucleotide anabolism, cellulose and mannose anabolism, and the glucagon signaling pathway (Fig 3E). GO analysis revealed that most of these genetic codes are involved in BPs, such as hydrogen peroxide reactions, drug reactions, diphosphate metabolism, hypoxia reactions, glucose metabolism, and cell division, as well as their molecular functions, including nucleoside diphosphate kinase activity, and kinase, monosaccharide, fatty acid, ATP, protein, RNA, and DNA binding (Fig 3F).

## 4.4. Establishment of stable cell line overexpressing ENO1

The fluorescence of SMMC-7721 cells transfected with *ENO1* overexpression lentivirus and SMMC-7721 cells transfected with empty lentivirus was 80%, whereas the fluorescence of SMMC-7721 cells transfected without *ENO1* overexpression lentivirus was 0%. After screening with puromycin at a concentration of 20 μg/μL, the *ENO1* overexpression transfected cells and empty vector-transfected cells grew well, whereas the untransfected cells were all killed. The *ENO1* expression in the transfected cells overexpressing *ENO1* at the transcription level was 25.09 times higher than that in the empty vector-transfected cells, indicating the establishment of stably transfected cells overexpressing *ENO1* (Fig 4).

## 4.5. Inhibitory effect of PP on hepatoma cells and its optimal concentration and time

The inhibitory effect of PP on stably transfected cells overexpressing *ENO1* increased in a dose-dependent manner. The logarithmic values of the concentration gradient of 1.25, 2.50, 5.00, 7.50,

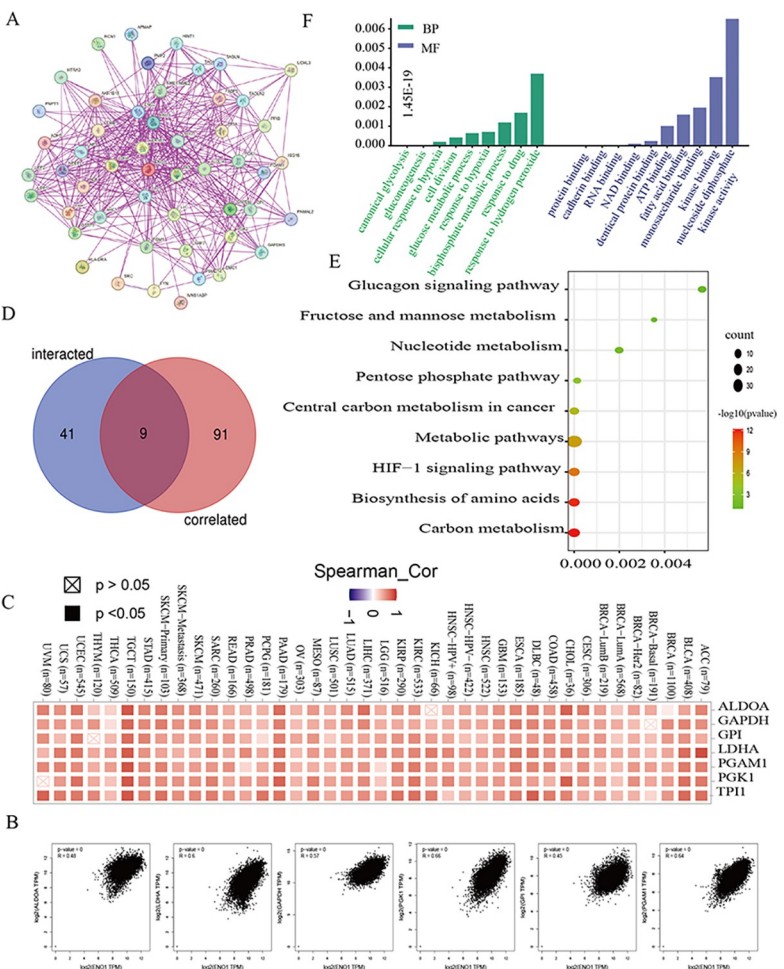

**Fig 3. Enrichment analysis of ENO1.** (A) 50 protein interaction networks related to ENO1; (B) The expression level of ENO1 was significantly correlated with glucose-6-phosphate isomerase (GPI), phosphoglycerate mutase 1 (PGAM1), LDHA, glyceraldehyde-3-phosphate dehydrogenase (GAPDH), fructose bisphosphate aldolase A (ALDOA) Phosphoglycerate kinase 1 (PGK1) and other genes were positively correlated ($P<0.001$); (C) Corresponding heatmap analysis shows that ENO1 is positively correlated with the six genes mentioned above. (D) There are 9 common genes interacting with and related to ENO1; (E) The role of ENO1 in tumor may be related to carbon metabolism, amino acid biosynthesis, HIF-1 signaling pathway, metabolic pathway, central carbon metabolism in cancer, pentose phosphate pathway, nucleotide metabolism, fructose and mannose metabolism, glucagon signaling pathway; (F) Most of these genes are involved in biological processes such as hydrogen peroxide reaction, drug reaction, diphosphate metabolic process, hypoxia reaction, glucose metabolic process, cell division, and have molecular functions such as nucleoside diphosphate kinase activity, kinase binding, monosaccharide binding, fatty acid binding, ATP binding, protein binding, RNA binding, DNA binding, etc.

and 10.00 μg/μL after 12 h of PP treatment exhibited an excellent linear relationship with the corresponding inhibition rate ($R^2 = 0.91$), with $IC_{50}$ was 5.78 μg/μL. However, after 24 h of treatment, the logarithm of the concentration gradient exhibited a poor linear relationship with the corresponding inhibition rate ($R^2 = 0.67$). Therefore, the concentration and action time of the PP solution in subsequent experiments were 5.78 μg/μL and 12 h, respectively (Fig 5).

## 4.6. PP inhibits the invasion of cells overexpressing ENO1

Overexpression of *ENO1* and empty transfection of hepatoma cells with PP solution reduced the number of invasive cells ($P < 0.01$). The number of invasive cells in the overexpression transfected hepatoma cells plus PP solution group was lower than that in the empty vector

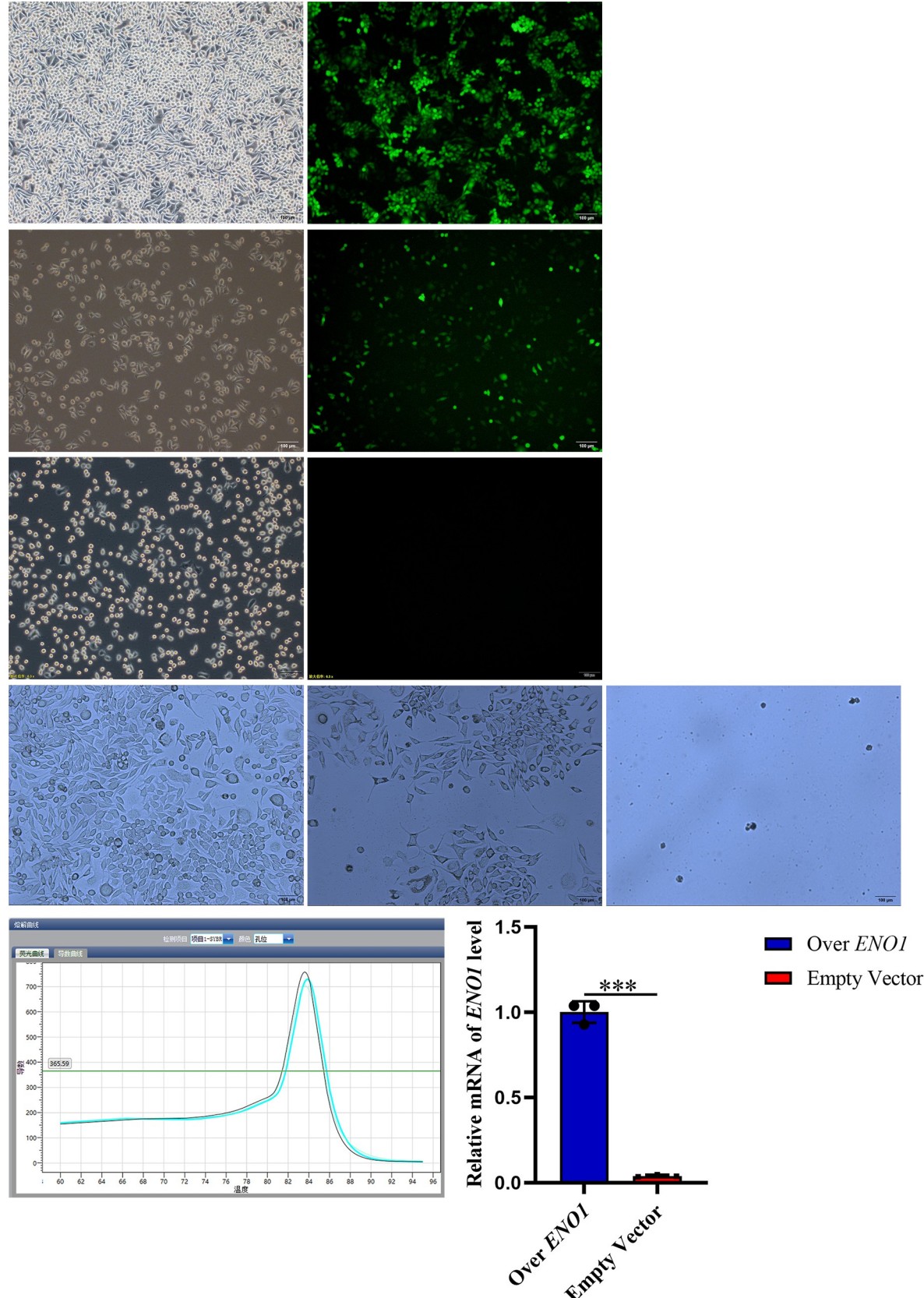

**Fig 4. In the transfected cells with over-expressed ENO1, ENO1 can be over-expressed.** (A) White light and fluorescence photos of overexpressed ENO1 cells transfected for 72 hours, with fluorescence cells accounting for approximately 80% of the white light cell count; Fluorescence represented successfully transfected cells. (B) White light and fluorescence photos of empty vector cells transfected for 72 hours, with fluorescent cells accounting for approximately 80% of the white light cell count; Fluorescence represented successfully transfected cells. (C) White light and fluorescence photos of non transfected cells cultured for 72 hours, but the cells did not show fluorescence; (D) Photos of overexpressed ENO1 cells screened with 20 µg/µl purinomycin for 3 days showed good cell growth; photos of empty transfected cells screened with 20 µg/µl purinomycin for 3 days showed good cell growth; and photos of non transfected cells screened with 20 µg/µl purinomycin for 3 days showed that all cells were killed by puromycin; Screening with puromycin involves killing untransfected cells that are not resistant to puromycin, while screening stable transfected cells that are resistant to puromycin. (E) RT-qPCR dissolution curve of overexpression ENO1 cells showed that the amplification product was ENO1; The histogram of relative expression of ENO1 RT-qPCR in overexpression ENO1 cells and empty vector cells showed the expression level of ENO1 in overexpressing ENO1 transfected cells was 25.09 times higher than that in empty transfected cells. These results indicated that ENO1 could be overexpressed in overexpressing ENO1 transfected cells.

cells without PP solution group ($P < 0.01$). These results indicated that PP could inhibit the invasion of stably transfected cells overexpressing *ENO1*. Furthermore, PP exhibited a more significant inhibitory effect on cells overexpressing *ENO1* than on empty vector cells, indicating that PP may act on *ENO1* to inhibit the invasion of liver cancer cells Table 1 and (Fig 6).

## 4.7. PP inhibits the migration of stably transformed cells overexpressing ENO1

The 24 h healing rate of the overexpressing *ENO1* cells plus PP group was lower than that of the empty transfected cells plus PP group ($P<0.05$) and empty vector cell group ($P<0.01$). The 48 h healing rate of cells overexpressing *ENO1* and treated with PP was lower than that of the empty transfected cells without the PP group ($P<0.05$). These results indicated that PP could inhibit the migration of cells overexpressing *ENO1*, but the inhibition of migration of empty-transfected cells was not evident Table 2 and (Fig 7).

## 4.8. Inhibition of ENO1 can effectively inhibit the proliferation of liver cancer cells

Compared with the control group without inhibitors, 0.2% of the DMSO used to dissolve inhibitors exhibited no inhibitory effect on stably transfected cells overexpressing *ENO1* and empty-transfected cells. Therefore, the effects on these two cell types were excluded. The survival rate of cells incubated with the ENO1 inhibitor decreased in a time- and dose-dependent manner compared to that of empty transfected cells without inhibitors (all $P < 0.05$), implying that *ENO1* inhibition can inhibit hepatoma cell proliferation (Fig 8).

**Table 1. Effect of PP on the invasive number of overexpressed and empty vector transformed hepatoma cells ($\bar{x}\pm s$), n = 3.**

| grouping | Number of cell invasion |
|---|---|
| Over *ENO1*+PP | 147.000±14.422**, ## |
| Over *ENO1* | 276.200±44.093 |
| Empty Vector+PP | 180.400±20.244## |
| Empty Vector | 266.8±39.124 |
| *F* | 19.910 |
| *P* | <0.001 |

Note

** indicates *P*< 0.01 compared with the overexpression *ENO1* without PP group

## means *P*< 0.01 Compared with empty vector group.

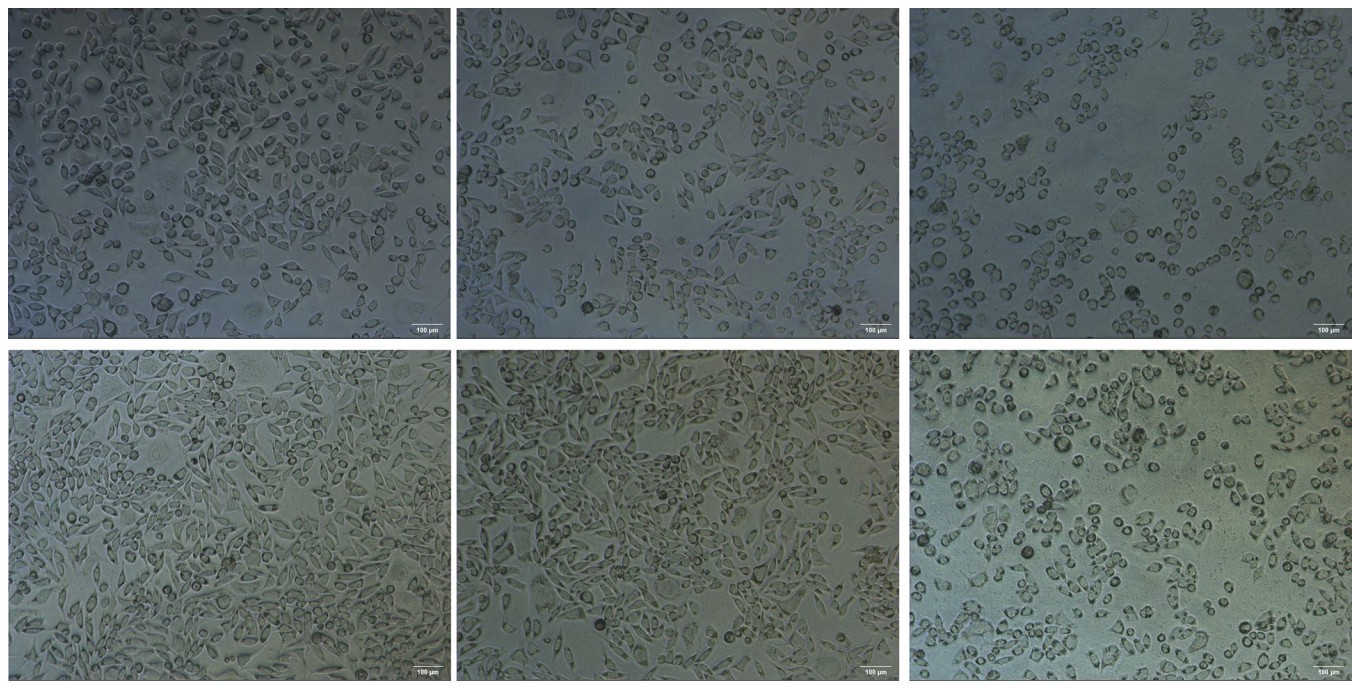

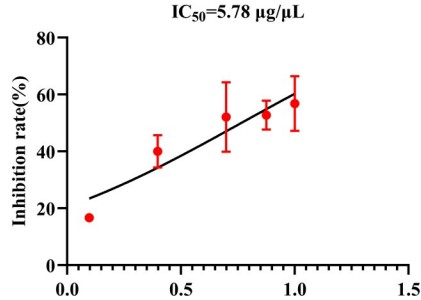

$IC_{50}=5.78 \mu g/\mu L$

**Fig 5. The optimal concentration and duration of PP's inhibitory effect on liver cancer cells.** (A) The empty vector transfected cells were cultured without PP solution for 12 hours as a negative control, and the overexpression ENO1 stable transfected cells was added with 5 μl PP solution hole culture for 12 hours and overexpression ENO1 stably transfected cells plus 75 μl PP solution holes were cultured for 12 hours; (B) The empty vector transfected cells were cultured for 24 hours without adding PP solution as a negative control, and the overexpression ENO1 stable transfected cells was added with 5 μl PP solution hole culture for 24 hours and overexpression ENO1 stably transfected cells plus 75 μl PP solution holes were cultured for 24 hours; These results indicated that, compared with the negative control, with the increase of the concentration of PP solution, the degree of inhibition of the overexpressed stable cells was more serious. (C) Relationship between the logarithmic value of drug concentration and corresponding cell inhibition rate after 12 hours of PP solution treatment. By calculation, IC50 = 5.78 μg/μl was obtained as the PP solution concentration and corresponding action time for subsequent experiments.

**Table 2. Effect of different treatment groups of SMMC-7721 transformed cells on their scratch healing rate($\bar{x}\pm s$), $n = 6$.**

| grouping | Over *ENO1*+PP | Over *ENO1* | Empty Vector+PP | Empty Vector | Z | P |
|---|---|---|---|---|---|---|
| 12h | 0.1274±0.0495 | 0.1579±0.0425 | 0.1151±0.0145 | 0.1297±0.0174 | 5.420 | 0.144 |
| 24h | 0.2113±0.0132*,## | 0.2890±0.0445 | 0.3421±0.1295 | 0.3205±0.0280 | 12.753 | 0.005 |
| 48h | 0.4153±0.0696# | 0.4684±0.0861 | 0.5025±0.1970 | 0.6902±0.1107 | 9.253 | 0.026 |
| F | 72.000 | | | | | |
| P | P<0.001 | | | | | |

Note

* means $P<0.05$ compared with Empty Vector+PP group

#, ## means $P<0.05$, $P<0.01$ compared with the Empty vector group, and the difference is statistically significant.

**Table 3. ALT, AST, AST/ALT and tumor weight of nude mice in different treatment groups ($\bar{x}\pm s$), $n = 6$.**

| group | ALT (U/L) | AST (U/L) | AST/ALT | Tumor weight (mg) |
|---|---|---|---|---|
| Over *ENO1*+PP | 372.5±30.7***, ### | 588.0±46.9***, ### | 1.58±0.02***, ## | 86.4±8.8***, ### |
| Over *ENO1*+PS | 529.8±34.5 | 1011.0±60.8 | 1.91±0.02 | 251.2±21.7 |
| Empty Vector+PP | 96.4±17.7 | 224.4±20.6 | 2.36±0.23 | 57.6±6.8 |
| Blank | 91.1±16.2 | 271.1±22.3 | 3.02±0.29 | —— |

Note

*** indicates that compared with overexpression *ENO1* plus PS group, $P < 0.001$, respectively.

##, ###, indicates compared with the empty vector plus PP group, $P < 0.01$, $P < 0.001$, respectively; "——" indicates that no tumor has grown.

## 4.9. PP reverses liver injury and tumor growth in mice with tumor overexpressing ENO1

The weight of the tumor was higher in the PS injection group, and the serum ALT, AST, and AST/ALT were significantly lower than those in the PP injection group and empty transfected cell injection group Table 3. However, there was no significant difference between the empty-transfected cells treated with the PP solution and the blank group. These data indicated that PP inhibited the proliferation of hepatoma cells by inhibiting ENO1 *in vivo*.

## 4.10. PP inhibits the expression of ENO1 and proliferation-related proteins in HCC cells in vitro and in vivo

*In vitro* experiments revealed lower CyclinE1, Erb-B2 receptor tyrosine kinase 2 (ERBB2), eukaryotic translation initiation factor2α (EIF2α), and ENO1 expression in the overexpression plus PP groups than in the respective empty vector without PP groups. Conversely, the overexpression plus PP and empty vector plus PP groups exhibited lower expression than the empty vector without PP group. The expression of AKT in the overexpression plus PP and empty vector plus PP groups was lower than that in the non-PP group. These results indicate that PP inhibited the expression of these proteins in hepatoma cells Tables 4 and 5 and Fig 9A and 9B.

*In vivo* experiments indicated that the protein expression of AKT in the empty-transfected cells plus PP group was significantly lower than in the overexpressed *ENO1* stable transfected cells plus PP and PS groups. The protein expression of C-MYC, PI3K, ERBB2, ENO1,

**Table 4. Comparison of CCNE1 and ERBB2 protein expression between experimental group and control group ($\bar{x}\pm s$), $n = 3$ (GAPDH have been removed from the value).**

| grouping | CCNE1 | ERBB2 |
|---|---|---|
| Over *ENO1*+PP | 0.462±0.004***, ### | 0.711±0.007***, ###, *** |
| Over *ENO1* | 0.596±0.018 | 0.784±0.021### |
| Empty Vector+PP | 0.483±0.007### | 0.439±0.005### |
| Empty Vector | 0.570±0.023 | 0.573±0.021 |
| *F* | 55.779 | 297.565 |
| *P* | <0.001 | <0.001 |

Note

*** indicates $P < 0.001$ compared with overexpression *ENO1* group

### indicates that compared with the empty vector group, $P < 0.001$; *** said compared with empty vector plus PP group, $P < 0.001$.

**Table 5. Comparison of AKT, ENO1and EIF2 α Protein expression between experimental group and control group ($\bar{x}\pm s$), $n$ = 3 (GAPDH have been removed from the value).**

| grouping | AKT | ENO1 | EIF2 α |
|---|---|---|---|
| Over *ENO1*+PP | 0.670±0.167* | 0.643±0.073### | 0.477±0.041***, ### |
| Over *ENO1* | 0.978±0.130 | 0.836±0.076 | 0.759±0.040# |
| Empty Vector+PP | 0.544±0.013# | 0.680±0.077### | 0.438±0.039### |
| Empty Vector | 0.811±0.158 | 1.093±0.132 | 0.839±0.042 |
| *F* | 5.975 | 14.671 | 74.295 |
| *P* | 0.019 | 0.001 | <0.001 |

Note

*, *** indicates $P<0.05$, $P<0.001$ compared with overexpression *ENO1* group, respectively

#, ##, ### indicate $P<0.05$, $P<0.01$, $P<0.001$, respectively, when compared with the empty vector group.

CyclinE1, and matrix metallopeptidase 2 (MMP2) in the empty load plus PP group was significantly lower than that in the overexpression plus PP and PS groups, and that in the overexpression plus PP group was significantly lower than that in the PS group. This indicates that *ENO1* is associated with the expression of other proteins, and PP inhibits the expression of ENO1-related proteins, except for AKT, *in vivo* Tables 6–8 and Fig 9C and 9D.

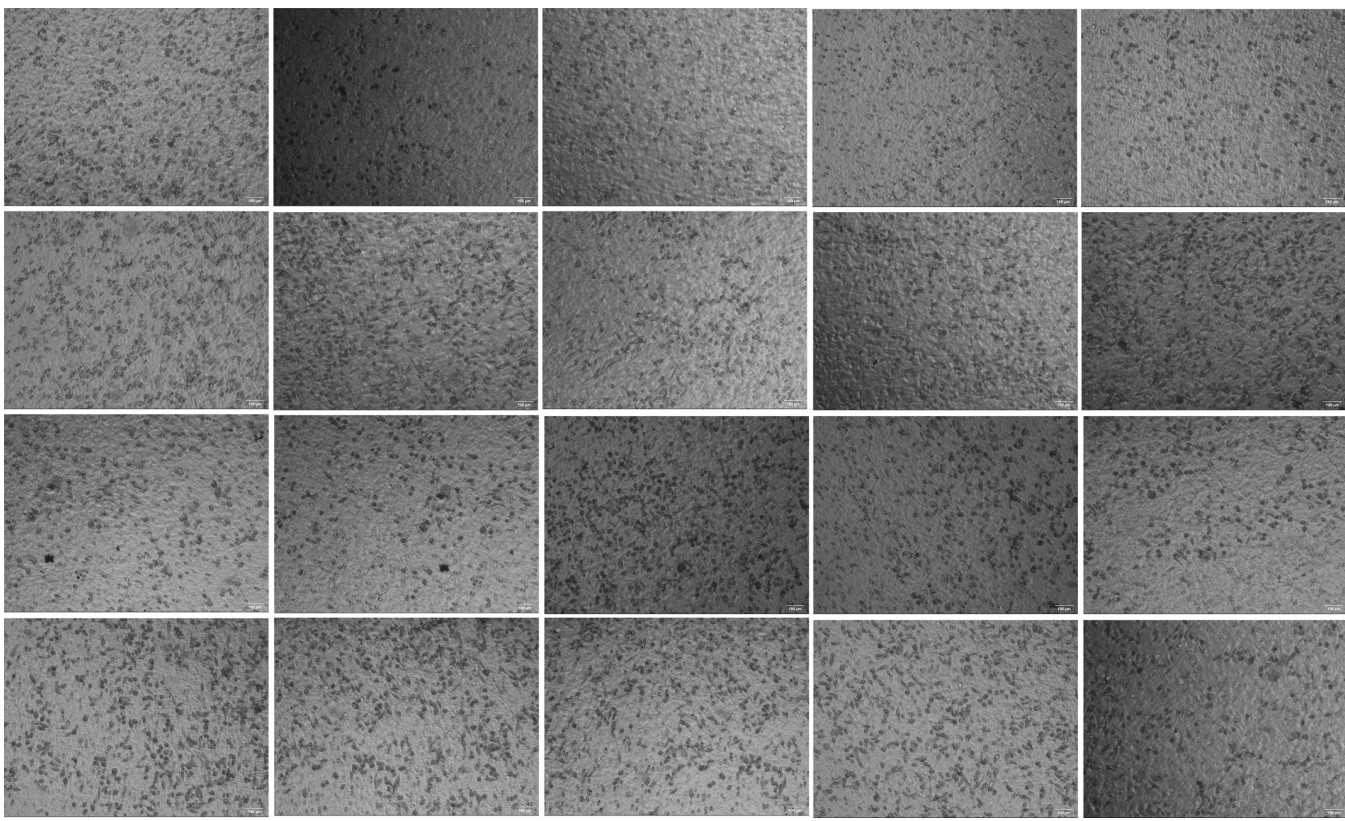

**Fig 6. Cell invasion photos of stable transfected cells overexpressing ENO1 and empty vector transfected cells.** (A) The stable transfected cells overexpressing ENO1 were treated with PP; (B) The stable transfected cells overexpressing ENO1 were not treated with PP; (C) Empty vector transfected cells plus PP group; (D) Empty vector transfected cells without PP group.

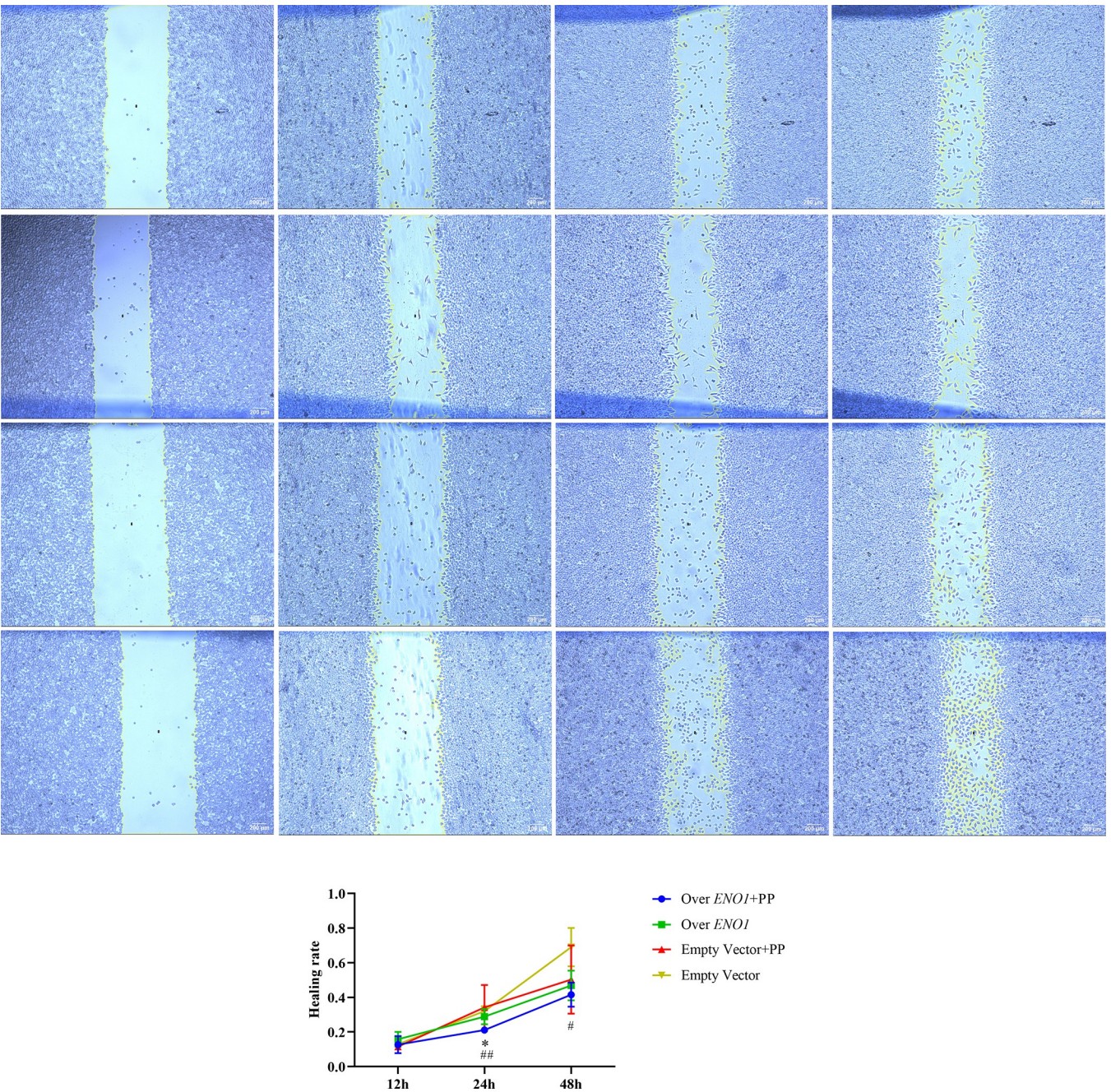

**Fig 7. PP inhibits the migration of ENO1 overexpressing stable transformed cells.** (A) Cell scratch of ENO1 overexpressing cells plus PP group at 0, 12, 24, and 48 hours; (B) Scratch of ENO1 overexpressing cells without PP group at 0, 12, 24, and 48 hours; (C) Cell scratch of empty vector transfected cells plus PP group at 0, 12, 24, and 48 hours; (D) Cell scratch of empty vector transfected cells without PP group at 0, 12, 24, and 48 hours; (E) PP inhibited the migration of ENO1 overexpressing cells; * means $P<0.05$ compared with Empty Vector + PP group, #, ## means $P<0.05$, $P<0.01$ compared with the Empty vector group, and the difference is statistically significant.

## 5. Discussion

ENO1 is a metal enzyme that catalyzes 2-phosphoglyceric acid to produce phosphoenolpyruvic acid during glycolysis [8]. C-MYC promoter-binding protein-1 (MBP-1) is another expression product of the *ENO1* gene that can inhibit tumor cells [9]. *ENO1* is highly expressed when the

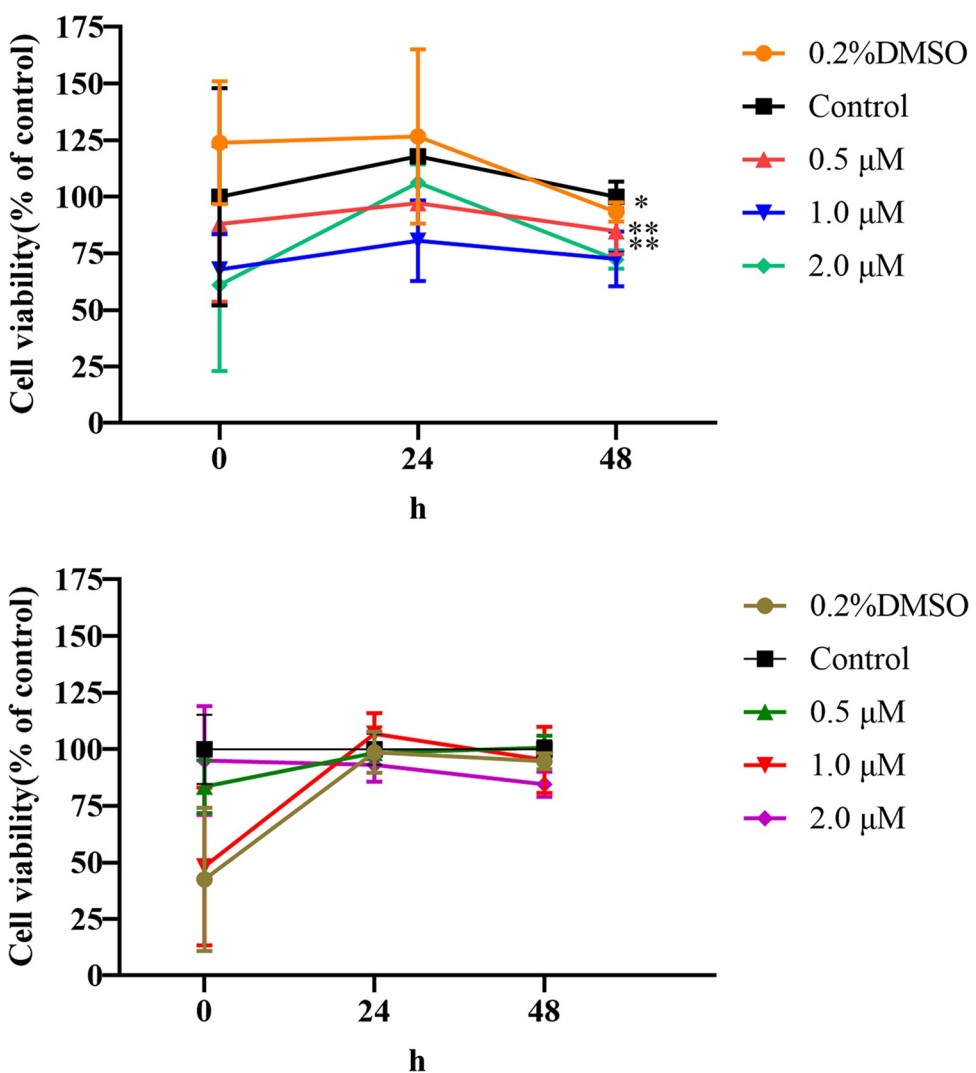

**Fig 8. Line plots of cell survival of ENO1-overexpressed stable transmutation cells and empty transfected cells.**
(A) The broken line diagram of the survival rate of the stably transfected cells expressing ENO1 under the influence of inhibitors with different concentrations and time of action; (B) the broken line diagram of the survival rate of the empty vector transfected cells under the influence of inhibitors of different concentrations and action time.

**Table 6. Comparison of AKT and C-MYC protein expression in different groups of cells ($\bar{x}\pm s$), $n$ = 3 (GAPDH have been removed from the value).**

| grouping | AKT | C-MYC |
|---|---|---|
| Over *ENO1*+PS | 0.918±0.069** | 0.555±0.067** |
| Over *ENO1*+PP | 0.830±0.058** | 0.413±0.058#* |
| Empty Vector+PP | 0.620±0.071 | 0.265±0.058 |
| *F* | 15.915 | 16.923 |
| *P* | 0.004 | 0.003 |

Note

*, ** means *P*< 0.05, *P*< 0.01 compared with empty vector plus PP group, respectively

# indicates *P*< 0.05 compared with the overexpression *ENO1* plus PS group.

**Table 7. Comparison of ERBB2, PI3K and CyclinE1 protein expression in different groups of cells ($\bar{x}\pm s$), $n$ = 3 (GAPDH have been removed from the value).**

| grouping | ERBB2 | PI3K | CyclinE1 |
|---|---|---|---|
| Over *ENO1*+PS | 0.811±0.069*** | 0.713±0.069** | 0.737±0.050*** |
| Over *ENO1*+PP | 0.632±0.072#*** | 0.556±0.058#* | 0.614±0.045#*** |
| Empty Vector+PP | 0.385±0.051 | 0.414±0.053 | 0.351±0.040 |
| *F* | 32.828 | 18.269 | 56.838 |
| *P* | 0.001 | 0.003 | <0.001 |

Note

*, **, *** means *P*< 0.05, *P*< 0.01, *P*< 0.001 compared with empty vector plus PP group, respectively

# indicates *P*< 0.05 compared with the overexpression *ENO1* plus PS group.

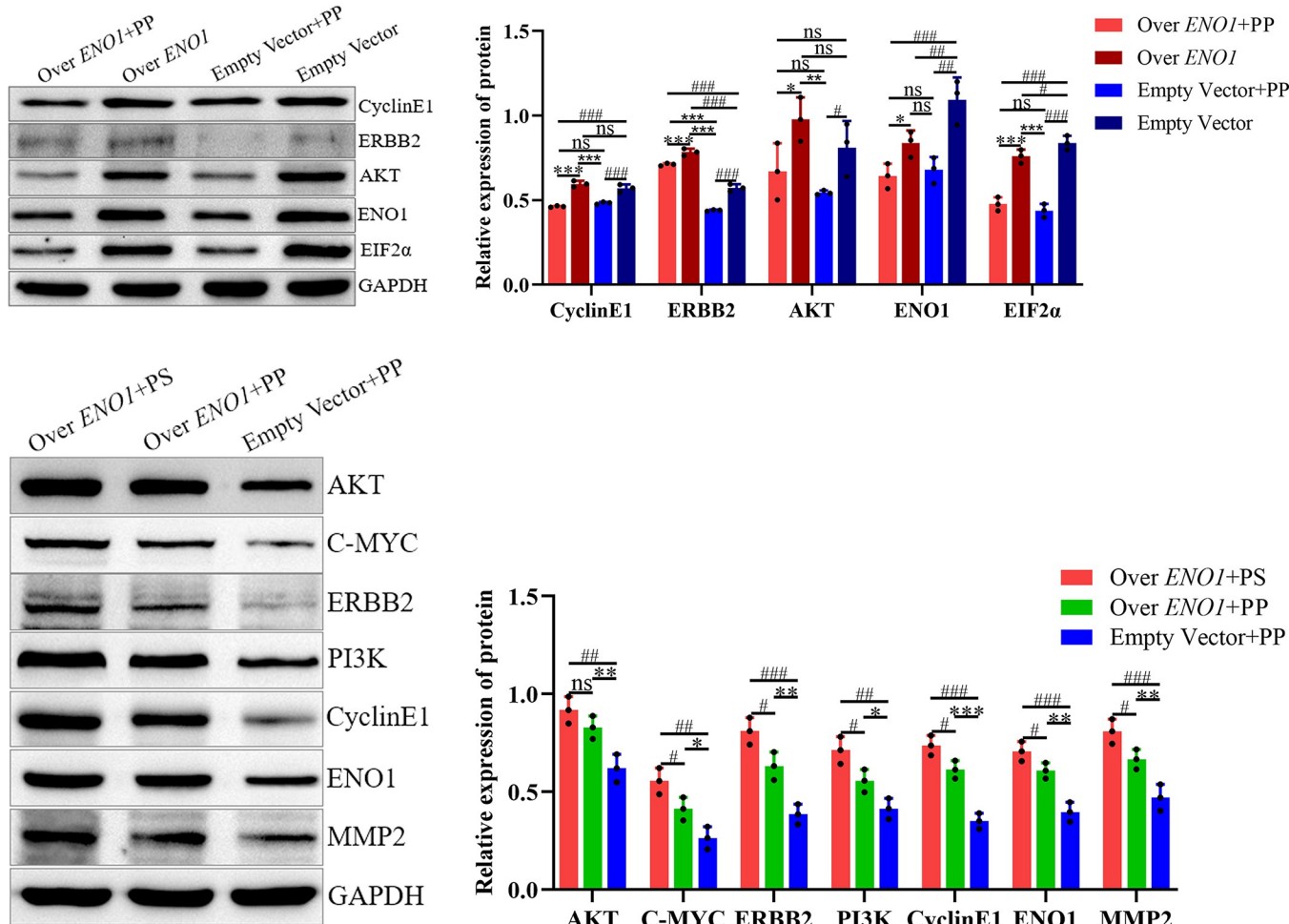

**Fig 9. Protein expression of AKT, C-MYC, ERBB2, PI3K, CyclinE1, ENO1, EIF2α and MMP2 in different groups.** (A) and (B) PP can inhibit CyclinE1, ERBB2, EIF2α and ENO1 expression in hepatoma cells in vitro; (C) And (D) PP could inhibit the expression of C-MYC, ERBB2, PI3K, CyclinE1, ENO1 and MMP2 in hepatoma cells in vivo; The expressions of AKT, C-MYC, ERBB2, PI3K, CyclinE1 and MMP2 in hepatocellular carcinoma cells were correlated with ENO1; Note: *: *P*< 0.05, **: *P*< 0.01, ***: *P*< 0.001; #: *P*< 0.05, ##: *P*< 0.01, ###: *P*< 0.001; **: *P*< 0.01, ***: *P*< 0.001.

**Table 8. Comparison of ENO1 and MMP2 protein expression in different groups of cells ($\bar{x}\pm s$), n = 3(GAPDH have been removed from the value).**

| grouping | ENO1 | MMP2 |
|---|---|---|
| Over *ENO1*+PS | 0.706±0.050*** | 0.809±0.063*** |
| Over *ENO1*+PP | 0.609±0.040#** | 0.666±0.050#** |
| Empty Vector+PP | 0.397±0.050 | 0.471±0.067 |
| *F* | 33.897 | 23.550 |
| *P* | 0.001 | 0.001 |

Note

**,*** means *P*< 0.01, *P*< 0.001 compared with empty vector plus PP group, respectively

# indicates *P*< 0.05 compared with the overexpression *ENO1* plus PS group.

tumor is at an advanced stage. A previous study reported that SMMC-7721 cell proliferation was achieved via the PI3K/AKT signaling pathway, which activates the upregulation of C-MYC [10, 11]. Moreover, ENO1 promotes the occurrence of various cancers, including breast cancer, through the PI3K/AKT pathway, and the PI3K/AKT pathway is also inhibited in liver cancer cells after inhibiting *ENO1* [12]. AKT inactivation can induce EIF2α expression. This activity was inhibited by increasing the expression of MBP-1, and MBP-1 was targeted to inhibit C-MYC expression [8, 13]. C-MYC also reportedly promotes *ENO1* expression and glycolysis [14]. Accordingly, the expression of *ENO1* and *MBP-1* exhibits a trade-off relationship.

We used the CPTAC database to explore the phosphorylation sites and ENO1 protein levels in Pan-cancer. Compared with normal tissues, the phosphorylation of ENO1 at the S27 site was the most significant in HCC and was closely related to the PI3K/AKT pathway. We also conducted a series of enrichment analyses of *ENO1* targeted binding protein and related genes. We identified that the role of ENO1 in tumor pathogenesis may involve BPs, such as hydrogen peroxide reaction, HIF-1 signaling pathway, glucose metabolism, and cell division [15].

Previous studies have reported that PP effectively inhibits the proliferation of HepG2 hepatoma cells [16]. In the present study, SMMC-7721 hepatoma cells were transfected to overexpress *ENO1*. The CCK-8 assay was used to determine the optimal concentration and action time of the PP solution, which was used as the action condition for PP in subsequent experiments. PP can inhibit the invasion of cells overexpressing *ENO1* by inhibiting the PI3K/AKT pathway and vascular endothelial growth factor expression related to epithelial-mesenchymal transition (EMT) [17]. Moreover, PP demonstrated a stronger inhibitory effect on the stable transformation of overexpressing *ENO1*, which suggests that PP may inhibit the invasion of hepatocellular carcinoma cells by acting on ENO1. It is conjectured that ENO1 acts as an upstream molecule in the PI3K/AKT pathway, and suppressing ENO1 can reduce the activation of the PI3K/AKT pathway and hinder the growth and spread of SK-Hep-1 liver cancer cells. [18], verifying this hypothesis. The cell scratch test further suggested that PP inhibited the migration of the stably transfected strain overexpressing *ENO1*, and this effect was linked to ENO1. In addition, studies have revealed that the PI3K/AKT pathway is closely linked to EMT in HepG2 and Huh7 liver cancer cells and that EMT is correlated with the migration of liver cancer cells [19]. Therefore, it is speculated that PP can hinder the migration of SMMC-7721 hepatoma cells overexpressing *ENO1* by inhibiting the ENO1 and PI3K/AKT pathways and inhibiting the EMT of HCC.

Subsequently, we investigated the signaling pathways and related protein expression. It has been reported that ERBB2 can activate the PI3K/AKT pathway and C-MYC, which is activated

by this pathway and upregulates *ENO1* expression. This results in a positive feedback cycle of repeated activation of the PI3K/AKT pathway, which in turn increases the production of vascular endothelial growth factor, MMP2, and other molecules related to the promotion of EMT, invasion, and migration of liver cancer cells [10, 13, 19–24]. Western blot analysis revealed that CyclinE1, ERBB2, and EIF2α were expressed in the liver cancer cells. PP inhibits the protein expression of ENO1 and AKT, and the production of CyclinE1 downstream of the PI3K/AKT pathway is associated with the growth and movement of liver cancer cells [25, 26]. In various cell lines, EIF2α endoplasmic reticulum stress causes inactivation of the PI3K/AKT pathway and is inhibited, thus increasing MBP-1 expression [13]. EIF2α, which is phosphorylated and inhibited by flavonoids, can cause apoptosis in liver cancer cells and inhibit HCC [27]. MBP-1 specifically inhibits ERBB2 expression [28]. The molecular mechanism that underpins the Transwell and cell scratch test results is elucidated by protein expression in this experiment and is consistent with the above research results. The main components of PP-containing flavonoids may inhibit the expression of ENO1 by enhancing the generation of MBP-1 and decreasing *ENO1* expression through the same mechanism. Subsequently, they can jointly prevent activation of the PI3K/AKT pathway, thus inhibiting the proliferation and migration of liver cancer cells.

The expressions of CyclinE1, ERBB2, and *ENO1* in the tumor were inhibited by PP *in vivo*, consistent with that observed *in vitro*. Furthermore, the expression levels of AKT, C-MYC, PI3K, ERBB2, ENO1, CyclinE1, and MMP2 in overexpressed *ENO1* cells with or without PP were significantly higher than those in empty cells with PP, indicating that the production rates of these proteins were positively associated with *ENO1*. The expression of these proteins is linked to the growth and movement of liver cancer cells.

To further illustrate the above inference, after inhibiting *ENO1* with different concentrations of inhibitors, the survival rate of cells overexpressing *ENO1* was significantly decreased, and cell proliferation was inhibited. In contrast, the degree of inhibition of empty-transfected cells expressing less *ENO1* was not obvious (Fig 8). This demonstrates that inhibition of *ENO1* can inhibit the proliferation of liver cancer cells from another aspect.

Furthermore, the size and mass of tumors treated with PP solution were significantly reduced compared to those treated with PS as a control. When the serum of each model group was analyzed, it was identified that among the results of ALT, AST, and AST/ALT, the level of *ENO1* overproduction in the PP group was less than that in the PS group. The increase in these biochemical indices was closely linked to the progression of HCC [29, 30]. This was consistent with the size and weight of each tumor.

## 6. Conclusion

In conclusion, our data suggest that PP can inhibit HCC by targeting the expression of *ENO1*, increasing *MBP-1* expression, and decreasing *ENO1* expression. *MBP-1* inhibits cancer, whereas *ENO1* has the opposite effect. Through targeted inhibition of C-MYC and ERBB2 by *MBP-1*, the positive feedback of PI3K/AKT activation is reversed, thereby forming the mechanism of inhibiting the proliferation and migration of liver cancer cells as a whole. Thus, the expression of proliferation- and migration-related proteins can be controlled, and an overall steady state can be achieved. This study offers a foundation for the identification and specialized treatment of HCC.

This study has several limitations. First, the inability to construct *an ENO1* knockdown lentivirus caused failure to perform relevant *in vitro* and *in vivo* experiments involving the knockdown *of ENO1*. Second, only SMMC-7721 cells were tested in this study. These data should be explored further in multiple cell lines.

## Supporting information

**S1 File. Raw images.** The raw images of Western blot.
(PDF)

**S2 File. Cell scratch test.** The cell scratch test statistical analysis software.
(RAR)

**S3 File. Data.** The software of test data.
(ZIP)

**S4 File. Fig 7E.** The drawing software of Fig 7E.
(RAR)

**S5 File. Table data.** The Original data in the table.
(DOCX)

## Acknowledgments

The authors thank Qun Wang and Lv Zhou for providing a way to polish the language.

## Author Contributions

**Conceptualization:** Yanhong Luo, Chun Guo, Weixin Xu.

**Data curation:** Yanhong Luo, Wenjun Yu, Chunfang Wang.

**Funding acquisition:** Chunfang Wang.

**Methodology:** Chun Guo, Caixia Ling, Weixin Xu.

**Software:** Yuanhong Chen, Lihe Jiang, Qiuxiang Luo.

**Writing – original draft:** Weixin Xu.

**Writing – review & editing:** Yanhong Luo, Chun Guo.

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
