## [Decision Letter · Decision Letter 0]

4 Jul 2024

PONE-D-24-17156Pine pollen reverses the function of hepatocellular carcinoma by inhibiting α-Enolase mediated PI3K / Akt signaling pathwayPLOS ONE

Dear Dr. Xu,

Thank you for submitting your manuscript to PLOS ONE. After careful consideration, we feel that it has merit but does not fully meet PLOS ONE’s publication criteria as it currently stands. Therefore, we invite you to submit a revised version of the manuscript that addresses the points raised during the review process.

We look forward to receiving your revised manuscript.

Kind regards,

Turki Talal Turki, Ph.D.

Academic Editor

PLOS ONE

Journal Requirements:

"This study was supported by the National Natural Science Foundation of China (No. 81960303); Project of Guangxi Key Laboratory of Molecular Pathology of Hepatobiliary Diseases (No. [2021]61); the Foundation of Modern Industrial College of Biomedicine and Great Health, Youjiang Medical University for Nationalities, Baise, Guangxi, China; School level project of Youjiang Medical University for Nationalities (No. yy2021sk012). "

"The authors confirm that they have no competing conflicts of interest."

5. In this instance it seems there may be acceptable restrictions in place that prevent the public sharing of your minimal data. However, in line with our goal of ensuring long-term data availability to all interested researchers, PLOS’ Data Policy states that authors cannot be the sole named individuals responsible for ensuring data access (http://journals.plos.org/plosone/s/data-availability#loc-acceptable-data-sharing-methods).

7. Your ethics statement should only appear in the Methods section of your manuscript. If your ethics statement is written in any section besides the Methods, please move it to the Methods section and delete it from any other section. Please ensure that your ethics statement is included in your manuscript, as the ethics statement entered into the online submission form will not be published alongside your manuscript. 

8. Please remove your figures from within your manuscript file, leaving only the individual TIFF/EPS image files, uploaded separately. These will be automatically included in the reviewers’ PDF."

Reviewers' comments:

Reviewer's Responses to Questions

**Comments to the Author**

1. Is the manuscript technically sound, and do the data support the conclusions?

Reviewer #1: Yes

Reviewer #2: Partly

2. Has the statistical analysis been performed appropriately and rigorously? 

Reviewer #1: Yes

Reviewer #2: No

3. Have the authors made all data underlying the findings in their manuscript fully available?

Reviewer #1: Yes

Reviewer #2: Yes

4. Is the manuscript presented in an intelligible fashion and written in standard English?

Reviewer #1: No

Reviewer #2: Yes

5. Review Comments to the Author

Reviewer #1: The manuscript entitled pollen reverses the function of hepatocellular carcinoma by inhibiting α-Enolase mediated PI3K PI3K/Akt pathway” pathway to  investigate the role of ENO1 in hepatocellular carcinoma cells and its relationship with the PI3K/Akt signalling signaling pathway. The focus was ENO1 ENO1, which highly expressed in tumour tumor and correlated with tumour tumor and metastasis. The Inhibiting ENO1 could reduce the of the PI3K/Akt signalling signaling pathway, inhibiting the proliferation and migration of hepatocellular carcinoma cells. The manuscript is well-organized and clearly stated. I would suggest accept accepting it the following major concerns are addressed:

1.The article discusses the role of a metalloenzyme, ENO1, in glycolysis and the regulation of its expression in tumor cells. The significance of ENO1 and the related research background could have been presented in more detail with more clarity in the introductory section.

2.The use of the CPTAC database to study the phosphorylation sites and levels of ENO1 in tumour tissues is mentioned in the article, and the methods and results of data analysis need to be explained in more detail to ensure the reliability and accuracy of the data.

3.Figure 6 is not clear and the need to avoid duplicates in Figure 6C.

4.Suggested supplementary tumour volume images in nude mice.

5.References need to be renewed. Some important recent studies regarding the hepatocellular carcinoma should be cited and discussed. For example, PMID: 38827325, 33987373, 38739668, etc.

6.It is suggested to add a supplementary figure with a graphic abstract which could better clarify the significance of this study.

Reviewer #2: Luo et al. presented a study applying various computational and experimental analysis to investigate the biological functions of Pine pollen in hepatocellular carcinoma. They found that Pine pollen inhibits HCC via ENO1, MBP-1 and PI3K/Akt pathway. The analysis and results appear intriguing and promising; however, there are several major issues in the current manuscript that undermine the study's clarity and conclusion.

1. The manuscript's English writing is quite poor, which significantly affects its readability. For example, on page 2 line 39, it is confusing to read “We analyzed the bioinformatics of ENO1”, which should be revised to “We applied various bioinformatic analysis of open-access data to study the expression of ENO1”. It is recommended that the authors seek assistance from a professional editing service or a native English speaker to enhance the quality of the writing.

2. The structure of the paper is currently disorganized, such that there are nine figures and seven tables totally, making it difficult for readers to follow the authors’ arguments and findings. The authors should reorganize the figures/tables to provide a coherent flow of information, such as putting some figures into supplementary file. Also, the index in Figure 3 is incorrect, please revise that.

3. The statistical analysis presented in the manuscript is not adequately justified and appears unreasonable in parts. For example, on page 18 line 184, the authors claimed that they applied one-way ANOVA test; however, at least in Figure 2 the test should be t-test or Wilcoxon test depends on the distribution of data. Besides, is the p-value in this study adjusted? The authors should revisit their analytical methods to ensure they are appropriate for their data.

4. The interpretation of the figures in the manuscript needs to be more accurate and detailed. Each figure and panel should be clearly explained. For example, how does each part in Figure 4 support the conclusion “Establishment of stable cell line overexpressing ENO1”?

Overall, the current manuscript cannot be considered for publication. The authors should improve the quality and clarity substantially so that the manuscript can meet the publication standards.

6. PLOS authors have the option to publish the peer review history of their article (what does this mean?). If published, this will include your full peer review and any attached files.

Reviewer #1: No

Reviewer #2: No

---

## [Author Response · Author response to Decision Letter 0]

14 Aug 2024

1. The English has been polished by authoritative institutions. 

2. The table of cell scratch experiment has been included as a supplementary document. The index of Figure 3 has been modified. 

3. The inter group analysis with time overlap adopts multiple analysis of variance. The numerical change was misread at the time, and the P-value has not been changed. I consulted with statistical experts, and they believe that this is correct.

4. The chart has been reorganized: Figure 4 (A) White light and fluorescence photos of overexpressed ENO1 cells transfected for 72 hours, with fluorescence cells accounting for approximately 80% of the white light cell count; Fluorescence represented successfully transfected cells. (B) White light and fluorescence photos of empty vector cells transfected for 72 hours, with fluorescent cells accounting for approximately 80% of the white light cell count; Fluorescence represented successfully transfected cells. (C) White light and fluorescence photos of non transfected cells cultured for 72 hours, but the cells did not show fluorescence; (D) Photos of overexpressed ENO1 cells screened with 20 μg/μl purinomycin for 3 days showed good cell growth; photos of empty transfected cells screened with 20 μg/μl purinomycin for 3 days showed good cell growth; and photos of non transfected cells screened with 20 μg/μl purinomycin for 3 days showed that all cells were killed by puromycin; Screening with puromycin involves killing untransfected cells that are not resistant to puromycin, while screening stable transfected cells that are resistant to puromycin. (E) RT-qPCR dissolution curve of overexpression ENO1 cells showed that the amplification product was ENO1; The histogram of relative expression of ENO1 RT-qPCR in overexpression ENO1 cells and empty vector cells showed the expression level of ENO1 in overexpressing ENO1 transfected cells was 25.09 times higher than that in empty transfected cells. These results indicated that ENO1 could be overexpressed in overexpressing ENO1 transfected cells.

Figure 5 (A) shows that the empty transfected cells were cultured in wells without PP solution for 12 hours as a negative control; Overexpression stable transformed cells were cultured in 5 μ L PP solution wells for 12 hours; Overexpression stable transformed cells were cultured in 75 μ L PP solution wells for 12 hours; (B) The empty transfected cells were cultured in wells without PP solution for 24 hours as a negative control; Overexpression stable transformed cells were cultured in 5 μ L PP solution wells for 24 hours; Overexpression stable transformed cells were cultured in 75 μ L PP solution wells for 24 hours. This indicates that compared to the negative control, as the concentration of PP solution increases, the degree of inhibition of overexpression stable transformed cells becomes more severe. (C) The relationship between the logarithm of drug concentration and the corresponding cell inhibition rate of PP solution after 12 hours of action was plotted. The calculated IC50 was 5.78 μ g/μ l, which was used as the PP solution concentration and corresponding action time for subsequent experiments.

---

## [Decision Letter · Decision Letter 1]

20 Sep 2024

PONE-D-24-17156R1Pine pollen reverses the function of hepatocellular carcinoma by inhibiting α-Enolase mediated PI3K/AKT signaling pathwayPLOS ONE

Dear Dr. Xu,

Thank you for submitting your manuscript to PLOS ONE. After careful consideration, we feel that it has merit but does not fully meet PLOS ONE’s publication criteria as it currently stands. Therefore, we invite you to submit a revised version of the manuscript that addresses the points raised during the review process.

We look forward to receiving your revised manuscript.

Kind regards,

Turki Talal Turki, Ph.D.

Academic Editor

PLOS ONE

Reviewers' comments:

Reviewer's Responses to Questions

**Comments to the Author**

1. If the authors have adequately addressed your comments raised in a previous round of review and you feel that this manuscript is now acceptable for publication, you may indicate that here to bypass the “Comments to the Author” section, enter your conflict of interest statement in the “Confidential to Editor” section, and submit your "Accept" recommendation.

Reviewer #1: (No Response)

Reviewer #2: (No Response)

2. Is the manuscript technically sound, and do the data support the conclusions?

Reviewer #1: (No Response)

Reviewer #2: Partly

3. Has the statistical analysis been performed appropriately and rigorously? 

Reviewer #1: (No Response)

Reviewer #2: N/A

4. Have the authors made all data underlying the findings in their manuscript fully available?

Reviewer #1: (No Response)

Reviewer #2: Yes

5. Is the manuscript presented in an intelligible fashion and written in standard English?

Reviewer #1: (No Response)

Reviewer #2: Yes

6. Review Comments to the Author

Reviewer #1: (No Response)

Reviewer #2: The authors addressed part of my previous comments and improved the manuscript to some extent. However, there are still some major issues with this paper:

Major comments:

1. When preparing the “Response To Reviewer Comments”, the authors should copy and paste the original comments from last round and then provide the point-by-point response to each comment. This is to maintain an easy and straightforward tracking of all comments and responses. Also, as the revision guidelines indicated, the authors should upload a marked-up copy of manuscript that highlights changes made to the original version (upload this as a separate file labeled 'Revised Manuscript with Track Changes')

2. “The English has been polished by authoritative institutions.” The English writing has indeed improved in current submission; However, there are still some sentences appear incorrect, such as in Abstract, Conclusion, the authors wrote “PP inhibits HCC by regulating the expression of ENO1 and MBP-1 and inhibiting the 53 PI3K/AKT pathway by inhibiting C-MYC and erb-B2 receptor tyrosine kinase 2.” It is confusing to read all three “inhibiting”, please revise that to make it clearer and more concise.

3. The format of reference numbers is incorrect. If the numbers are listed as superscript, then there should not be brackets around them. Please revise that and follow the official format by the journal.

7. PLOS authors have the option to publish the peer review history of their article (what does this mean?). If published, this will include your full peer review and any attached files.

Reviewer #1: No

Reviewer #2: No

---

## [Author Response · Author response to Decision Letter 1]

3 Oct 2024

1. The manuscript's English writing is quite poor, which significantly affects its readability. For example, on page 2 line 39, it is confusing to read “We analyzed the bioinformatics of ENO1”, which should be revised to “We applied various bioinformatic analysis of open-access data to study the expression of ENO1”. It is recommended that the authors seek assistance from a professional editing service or a native English speaker to enhance the quality of the writing.

Answer: The English has been polished by authoritative institutions. 

2. The structure of the paper is currently disorganized, such that there are nine figures and seven tables totally, making it difficult for readers to follow the authors’ arguments and findings. The authors should reorganize the figures/tables to provide a coherent flow of information, such as putting some figures into supplementary file. Also, the index in Figure 3 is incorrect, please revise that.

Answer: The table of cell scratch experiment has been included as a supplementary document. The index of Figure 3 has been modified. 

3. The statistical analysis presented in the manuscript is not adequately justified and appears unreasonable in parts. For example, on page 18 line 184, the authors claimed that they applied one-way ANOVA test; however, at least in Figure 2 the test should be t-test or Wilcoxon test depends on the distribution of data. Besides, is the p-value in this study adjusted? The authors should revisit their analytical methods to ensure they are appropriate for their data.

Answer: Multivariate analysis of variance is used for intergroup analysis with temporal overlap in statistics. The numerical change was misread at the time, and the P-value has not been changed. I consulted with statistical experts, and they believe that this is correct. 

4. The interpretation of the figures in the manuscript needs to be more accurate and detailed. Each figure and panel should be clearly explained. For example, how does each part in Figure 4 support the conclusion “Establishment of stable cell line overexpressing ENO1”?

Answer: The chart has been reorganized: Figure 4 and Figure 4 (A) show the white light and fluorescence images of cells overexpressing ENO1 after transfection for 72 hours, with fluorescent cells accounting for approximately 80% of the white light cells; Fluorescence represents successfully transfected cells. (B) White light and fluorescence photos of empty vector cells transfected for 72 hours, with fluorescent cells accounting for approximately 80% of the white light cells; Fluorescence represents successfully transfected cells. (C) White light and fluorescence photos of untransfected cells cultured for 72 hours, but the cells did not show fluorescence; (D) After screening cells overexpressing ENO1 with 20 μ g/μ l puromycin for 3 days, the photos showed good cell growth; After screening empty vector transfected cells with 20 μ g/μ l puromycin for 3 days, the photos showed good cell growth; After screening untransfected cells with 20 μ g/μ l puromycin for 3 days, the photos showed that all cells were killed by puromycin; Screening with puromycin involves killing untransfected cells that are intolerant to puromycin, while selecting stable transfected cells that are tolerant to puromycin. (E) The RT qPCR dissolution curve of overexpressing ENO1 cells showed that the amplified product was ENO1; The histogram of relative expression levels of ENO1 RT qPCR in overexpressing ENO1 cells and empty vector cells showed that the expression level of ENO1 in overexpressing ENO1 transfected cells was 25.09 times higher than that in empty vector transfected cells. These results indicate that ENO1 can be overexpressed in cells transfected with ENO1. 

Figure 5 (A) shows that the empty transfected cells were cultured in wells without PP water extract for 12 hours as a negative control; Overexpression stable transformed cells were cultured in 5 μ L PP water extract wells for 12 hours; Overexpression stable transformed cells were cultured in 75 μ L PP water extract wells for 12 hours; (B) The empty transfected cells were cultured in wells without PP water extract for 24 hours as a negative control; Overexpression stable transformed cells were cultured in 5 μ L PP water extract wells for 24 hours; Overexpression stable transformed cells were cultured in 75 μ L PP water extract wells for 24 hours. This indicates that compared to the negative control, as the concentration of PP water extract increases, the degree of inhibition of overexpression stable transformed cells becomes more severe. (C) The relationship between the logarithm of drug concentration and the corresponding cell inhibition rate of PP solution after 12 hours of action was plotted. The calculated IC50 was 5.78 μ g/μ l, which was used as the PP solution concentration and corresponding action time for subsequent experiments. 

Reviewer's comment response: 

1. When preparing the “Response To Reviewer Comments”, the authors should copy and paste the original comments from last round and then provide the point-by-point response to each comment. This is to maintain an easy and straightforward tracking of all comments and responses. Also, as the revision guidelines indicated, the authors should upload a marked-up copy of manuscript that highlights changes made to the original version (upload this as a separate file labeled 'Revised Manuscript with Track Changes')

Answer: We have responded to the comments one by one and provided a manuscript with revision marks, named "Revised Manuscript". 

2. “The English has been polished by authoritative institutions.” The Englishwriting has indeed improved in current submission; However, there are still some sentences appear incorrect, such as in Abstract, Conclusion, the authors wrote “PP inhibits HCC by regulating the expression of ENO1 and MBP-1 and inhibiting the 53 PI3K/AKT pathway by inhibiting C-MYC and erb-B2 receptor tyrosine kinase 2.” It is confusing to read all three “inhibiting”, please revise that to make it clearer and more concise.

Answer: The second "inhibiting" in "PP inhibitors HCC by regulating the expression of ENO1 and MBP-1 and inhibiting the PI3K/AKT pathway by inhibiting C-MYC and erb-B2 receptor tyrosine kinase 2." has been changed to "suppressing". 

3. The format of reference numbers is incorrect. If the numbers are listed as superscript, then there should not be brackets around them. Please revise that and follow the official format by the journal.

Answer: The superscript of the reference number has been changed to normal.

---

## [Decision Letter · Decision Letter 2]

8 Oct 2024

Pine pollen reverses the function of hepatocellular carcinoma by inhibiting α-Enolase mediated PI3K/AKT signaling pathway

PONE-D-24-17156R2

Dear Dr. Xu,

We’re pleased to inform you that your manuscript has been judged scientifically suitable for publication and will be formally accepted for publication once it meets all outstanding technical requirements.

Kind regards,

Turki Talal Turki, Ph.D.

Academic Editor

PLOS ONE

Additional Editor Comments (optional):

Reviewers' comments:

Reviewer's Responses to Questions

**Comments to the Author**

1. If the authors have adequately addressed your comments raised in a previous round of review and you feel that this manuscript is now acceptable for publication, you may indicate that here to bypass the “Comments to the Author” section, enter your conflict of interest statement in the “Confidential to Editor” section, and submit your "Accept" recommendation.

Reviewer #2: All comments have been addressed

2. Is the manuscript technically sound, and do the data support the conclusions?

Reviewer #2: Yes

3. Has the statistical analysis been performed appropriately and rigorously? 

Reviewer #2: N/A

4. Have the authors made all data underlying the findings in their manuscript fully available?

Reviewer #2: Yes

5. Is the manuscript presented in an intelligible fashion and written in standard English?

Reviewer #2: Yes

6. Review Comments to the Author

Reviewer #2: All comments have been addressed in this revised manuscript, and it has been improved to a large extent.

7. PLOS authors have the option to publish the peer review history of their article (what does this mean?). If published, this will include your full peer review and any attached files.

Reviewer #2: No

---

## [Editor Report · Acceptance letter]

4 Nov 2024

PONE-D-24-17156R2 

PLOS ONE

Dear Dr. Xu, 

I'm pleased to inform you that your manuscript has been deemed suitable for publication in PLOS ONE. Congratulations! Your manuscript is now being handed over to our production team.

Kind regards, 

on behalf of

Dr. Turki Talal Turki 

Academic Editor

PLOS ONE